# A Coordinated Vehicle–Drone Arc Routing Approach Based on Improved Adaptive Large Neighborhood Search

**DOI:** 10.3390/s22103702

**Published:** 2022-05-12

**Authors:** Guohua Wu, Kexin Zhao, Jiaqi Cheng, Manhao Ma

**Affiliations:** 1School of Traffic & Transportation Engineering, Central South University, Changsha 410075, China; guohuawu@csu.edu.cn (G.W.); kx_zhao@csu.edu.cn (K.Z.); 204207032@csu.edu.cn (J.C.); 2College of Systems Engineering, National University of Defense Technology, Changsha 410073, China

**Keywords:** vehicle–drone, arc routing problem, traffic patrol, routing optimization, adaptive large neighborhood search

## Abstract

Through urban traffic patrols, problems such as traffic congestion and accidents can be found and dealt with in time to maintain the stability of the urban traffic system. The most common way to patrol is using ground vehicles, which may be inflexible and inefficient. The vehicle–drone coordination maximizes utilizing the flexibility of drones and addresses their limited battery capacity issue. This paper studied a vehicle–drone arc routing problem (VD-ARP), consisting of one vehicle and multiple drones. Considering the coordination mode and constraints of the vehicle–drone system, a mathematical model of VD-ARP that minimized the total patrol time was constructed. To solve this problem, an improved, adaptive, large neighborhood search algorithm (IALNS) was proposed. First, the initial route planning scheme was generated by the heuristic rule of “Drone-First, Vehicle-Then”. Then, several problem-based neighborhood search strategies were embedded into the improved, adaptive, large neighborhood search framework to improve the quality of the solution. The superiority of IALNS is verified by numerical experiments on instances with different scales. Several critical factors were tested to determine the effects of coordinated traffic patrol; an example based on a real road network verifies the feasibility and applicability of the algorithm.

## 1. Introduction

With the continuous increase in car ownership in urban areas, congestion, traffic accidents, and other adverse factors seriously affect the operation of the urban traffic system [1], which is a huge challenge for traffic management departments. In order to maintain the stability of the urban traffic system, a traffic patrol of the urban road network is essential. At present, there are problems such as poor pertinence, inflexibility, and untimely monitoring in traditional police patrols using ground vehicles. By developing civil drones and making full use of their flexibility, mobility, and timeliness, the efficiency of urban traffic patrols can be further improved [2]. However, the widespread application of drones is limited by their battery capacities [3].

The combination of vehicle and drone gives full range to their advantages, and makes up for their shortcomings [4]. Drones are less limited by ground traffic, and can perform better in places that are difficult for vehicles to reach. Meanwhile, when not visiting tasks, the drones stay on the vehicle and the battery can be replaced to prepare for the next flight. In the past few years, research on coordinated vehicle–drone routing was mostly applied to parcel deliveries [5]. For example, Murray and Chu [4] first proposed the flying sidekick traveling salesman problem (FSTSP) in 2015, to study the parcel delivery problem considering a vehicle and a drone. In this problem, the drone can only visit one customer after launching from the vehicle each time, and the vehicle cannot stop in place to wait and recover the drone. Agatz et al. [6] propose the traveling salesman problem with drone (TSP-D), and introduce a heuristic approach to solve it. Compared with FSTSP, in the TSP-D, the drone can be launched and recovered at the same node. Based on the above two studies, a large number of studies emerged considering more complex constraints, such as time windows [7], different objective functions [8,9], heterogeneous drones [10], and simultaneous pickup and delivery [11]. Meanwhile, heuristic methods [6,12], a simulated annealing algorithm [13], a genetic algorithm [14], and variable neighborhood search algorithm [15] were designed to solve these problems. However, although some of the above studies on coordinated vehicle-drone parcel delivery aim at visiting customer nodes, they are still variants of the vehicle routing problem (VRP) [16]. This paper studied patrol routing of the urban road network, and focused on segments of the road network, that is, the road network itself. This problem can be defined as an extension of the vehicle–drone arc routing problem [17].

The vehicle–drone arc routing problem can be applied to many scenarios, such as urban traffic patrol and road snow clearing, and has great potential for research and application [18]. However, there are few studies on arc routing of vehicle–drone coordination. Among the research available, there are only two studies on the coordinated vehicle–drone arc routing problem. Liu et al. [19] apply the combination mode of a vehicle and a drone to high-voltage powerline inspection, in which the drone inspects the powerline and the vehicle is used for launching/recovering the drone at a specific rendezvous. Vehicle and drone traveled on different maps, where the planning of vehicle path is still a VRP. Luo et al. [20] first propose and solve a traffic patrol model with a vehicle and a drone, and they consider both arc tasks and point tasks. However, the above two studies only consider patrolling with the combination of one vehicle and one drone, which is inefficient compared with that of one vehicle and multiple drones. Hence, this study focused on a more difficult problem, which allows one vehicle and multiple drones to patrol cooperatively. Besides, to be more realistic, we consider that the drones fly without following the road network. We name the studied problem the vehicle–drone arc routing problem, consisting of a vehicle and multiple drones (VD-ARP).

It is a great challenge to solve VD-ARP and give the routing planning scheme of the vehicle and the drones. First, considering the traffic patrol requirements, the vehicle must move along the road network, and needs to be coordinated with multiple drones, which makes it very difficult to handle the routing constraints. Second, the encoding method of the vehicle route and drone routes may affect the complexity of solving the problem, while it is a challenge to design an elegant encoding of the target edges. Third, it is worth considering how to design the search framework and effective neighborhood search strategies, so that the algorithm can meet the time and efficiency requirements of an urban traffic patrol. In this paper, all the above challenges were properly addressed. Specifically, an improved, adaptive, large neighborhood search algorithm (IALNS) was designed. IALNS searched the solution space and obtained new alternative solutions through the adaptive large neighborhood search mechanism, and received the new solution, according to the Metropolis criterion in the simulated annealing algorithm. Moreover, a tabu table was set up in the search process, in order to reduce the repeated search in a short period of time.

The major contributions of this paper are as follows.
We proposed a vehicle–drone arc routing problem considering a vehicle and multiple drones (VD-ARP), and considered a more realistic scenario in which drones fly outside of the road network in addition to visiting tasks. This was rarely investigated in the past.To solve VD-ARP, we designed an improved, adaptive, large neighborhood search algorithm (IALNS). First, the task edges were encoded and the initial solution was generated by the “Drone-First, Vehicle-Then” heuristic method. Then, a set of problem-based neighborhood search strategies were designed. Finally, the simulated annealing mechanism, tabu table, and multiple neighborhood search strategies were embedded into the adaptive large neighborhood search framework to improve the solution. Compared with the existing heuristic methods, IALNS can be applied to larger-scale tasks, and obtain an acceptable solution in a short time.The effectiveness and superiority of IALNS are verified based on the experiments of instances with different scales. In addition, a case study was carried out to show that IALNS provides a satisfactory traffic patrol routing scheme in a short time.

The remainder of this paper is structured as follows. Section 2 provides a brief review of related works and Section 3 describes the VD-ARP and gives a mathematical model. Section 4 details the proposed improved, adaptive, large neighborhood search algorithm, and Section 5 reports and discusses the experimental studies. Finally, Section 6 gives conclusions.

## 2. Related Work

In recent years, traffic patrols have become an important way to prevent and respond to emergencies in urban traffic management [21]. Through traffic patrols, it is possible to intervene in congestion and monitor vehicle violations, as well as road facilities and equipment. Steil et al. [22] study the expression, execution, evaluation, and engagement of patrol routing algorithms for accident-prone locations on roads, so as to provide acceptable patrol routes for organizations such as police agencies and emergency medical responders. Jalili et al. [23] design a set of comprehensive measures for highway patrol agencies to measure and improve the management effectiveness, in order to improve road network safety. Li et al. [24] propose a new evaluation standard to evaluate the effect of traffic patrols in emergency situations. Chawathe et al. [25] solve a problem of patrol route planning, aiming at minimizing the length of patrol routes, and maximizing coverage of patrol locations. There are also some studies related to traffic patrols with drones. Wang et al. [26] use drones to collect data for vehicle speed analyses in the road network. Munishkin et al. [27] and Wu et al. [28] propose an information transmission model and an adaptive data processing system for vehicles and drones respectively, providing theoretical support for a coordinated traffic patrol of vehicles and drones. The main contribution of this paper is to design a route planning algorithm for a coordinated vehicle–drone traffic patrol problem by taking the required road segments as patrol targets.

When the patrol targets of vehicles and drones are segments in the road network, this problem is described as a variant of the arc routing problem [17]. The arc routing problem was developed from graph theory [29], and the most widely studied arc routing problems include the Chinese postman problem (CPP) [30,31], rural postman problem (RPP) [32,33,34], and capacitated arc routing problem (CARP) [35,36]. Kwan [37] first proposed the CPP, and designed a deterministic solution method. The problem is described as traversing all edges of a graph, and minimizing the total distance. Lima et al. [38] solve the CPP based on a mixed graph through ant colony algorithm, which is proven to be an NP-hard problem. As for RPP, only a part of the edges in the graph should be the accesses. Hertz et al. [39] reconstruct the undirected RPP problem, and design a variety of heuristic methods to obtain high-quality solutions. Fernandez et al. [40] study a multi-depot RPP, in which each route starts and ends at the same depot, and develop a branch-and-cut algorithm. Zhang et al. [41] propose a multi-depot CARP of a large scale, and introduce an iterative improved heuristic algorithm to solve it. Babaee et al. [42] design a hybrid ant colony optimization algorithm to solve the multi-trip CARP, and apply it to urban solid waste management. With the widespread use of drones, the arc routing problem has been extended, since drones can fly between any two nodes without following the road network. Oh et al. [43] present the Chinese postman problem with drones, traversing all edges of the road network by drones. Campbell et al. [44] first define the drone arc routing problem (DARP), in which the drone can move directly between any two points in the graph, and the entire edge accessed through multiple drone flights. However, the above studies on arc routing are all for the route planning of a patrol platform (vehicles or drones), while this paper considers both vehicles and drones, which is a more complex combinatorial optimization problem.

In the last few years, research related to route planning combining vehicles and drones gradually emerged, and was applied to parcel deliveries [45]. Murray and Chu [4] and Agatz et al. [6] first propose FSTSP and VRP-D, respectively, to enable a vehicle and a drone to visit customer nodes in parallel, and design heuristic methods to plan the vehicle and drone routes. In the literature review of Chung et al. [5], the research on various joint vehicle-drone operations, based on parcel delivery, is introduced in detail. However, to the best of our knowledge, there are few related studies on the arc routing problem for coordinated vehicle–drone operations. Luo et al. [20] study an urban patrol problem considering a vehicle and a drone, and design a heuristic method to solve the problem. However, in this paper, we consider the combination of multiple drones and one vehicle, and assume that the drones can fly between any two nodes in the road network.

Table 1 presents the research status of representative parts of relevant works. Obviously, most studies consider road network scenarios. As for the research on routing optimization, there are two types of tasks, i.e., arc and point. It should be noted that most of the studies on routing optimization are based on one platform (vehicles or drones), while this paper focuses on the coordination of vehicles and drones. Moreover, different from [20], we consider the combination of one vehicle and multiple drones, which is more complex and flexible than that of one vehicle and one drone.

## 3. Problem Description

The proposed VD-ARP in this paper can be described as: a vehicle departs from the depot, carrying multiple drones, which can be launched and recovered several times from the vehicle, and the vehicle and the drones jointly access all target edges in the road network. Finally, the vehicle returns to the depot with all the drones. Once a drone accesses the required target edges in a flight, it needs to return to the vehicle. The vehicle also acts as an energy supply station for the drones to replace the battery. In this problem, some of the target edges are required to be accessed only by drones. To simplify the problem and facilitate the model, we consider the following assumptions:The launch or recovery operations of a drone must be carried out at a node (intersection), and the vehicle can perform multiple launch/recovery operations at one node;After the vehicle launches/recovers a drone, it can wait in place, or go to another node to recover the drone or launch another drone;The vehicle must move along the road network, while the drone routes are not restricted by the road network;The launch/recovery time of a drone is very short compared to the travel time on the road. Therefore, as in references [6,19], the time of launching/recovering a drone is incorporated into the travel time here.

The urban road network is simplified as an undirected connected graph, represented by G=(V,E). The set of nodes (intersections of the road network) is expressed as V={1,2,…,n}, where n is the number of nodes. The set of edges (road segments) is denoted as $E=\left\{e_{ij}=\left(i,j\right)|i,j\in V\right\}$, $e_{ij}$ for connecting nodes i and j. The road network is undirected, so $e_{ij}$ and $e_{ji}$ represent the same edge. Since the vehicle moves along the road network, in order to distinguish the start nodes and end nodes of all edges that the vehicle passes through, the set $V_{-}$ represents the start node set, and $V_{+}$ represents the end node set. Depots in both sets are represented by $v_{0}$ and $v_{PC}$, respectively. T is the general target edge set, while TD is the target edge set that can only be accessed by drones, T,TD⊆E. Figure 1 is a schematic diagram of the road network and target edges. In the figure, according to the above definition, T={(13,21)(11,16)(15,16)} and TD={(7,8)(9,14)}. In addition, other symbols and descriptions are shown in Table 2.

As mentioned above, the objective function consists of two parts: the time it takes for the vehicle to arrive at the depot, and the time spent waiting at the depot for all drones to return to the vehicle. Hence, the optimization objective of the patrol routing planning problem is the total patrol time, which can be written as
(1)$$\min t=t_{PC}+\varepsilon_{PC}$$
where the first part is the time when the vehicle arrives at the depot $v_{PC}$, and $\varepsilon_{PC}$ represents the waiting time for the vehicle to recover all drones at the depot. When the vehicle is not the last to arrive at node i, $\varepsilon_{i}=\left(\max\left\{t_{i}^{k}\right\}-t_{i}\right)$; when the vehicle arrives last, $\varepsilon_{i}=0$. It is worth noting that $t_{i}^{k}$ denote the time when the k-th drone arrives at node i, where k∈D.

In addition, the main constraints that the problem should satisfy are as follows.
(2)$$\sum_{e_{i,k}\in E}x_{i,k}=\sum_{e_{k,j}\in E}x_{k,j},\forall k\in V$$
(3)$$\sum_{e_{0,i}\in E}x_{0,i}=1,\forall i\in V_{+}$$
(4)$$\sum_{e_{j,PC}\in E}x_{j,PC}=1,\forall j\in V_{-}$$
(5)$$x_{i,j}+x_{j,i}+\sum \left(y_{i,j}^{k,f}+y_{j,i}^{k,f}\right)\geq 1,\forall e_{ij}\in T$$
(6)$$\sum \left(y_{i,j}^{k,f}+y_{j,i}^{k,f}\right)\geq 1,\forall e_{ij}\in TD$$
(7)$$\sum_{j\in V}z_{i,j}^{k,f}=1,\;\forall i\in V,k\in D$$
(8)$$\sum_{i\in V}z_{i,j}^{k,f}=1,\forall j\in V,k\in D$$
(9)$$\sum_{j\in V}z_{i,j}^{k,f}\leq \sum_{s\in V}x_{is},\forall i,j\in V,j\neq s$$
(10)$$\sum_{i\in V}z_{i,j}^{k,f}\leq \sum_{v\in V}x_{vj},\forall i,j\in V,i\neq v$$
(11)$$av_{i}^{k}\left(\sum_{j\in V^{+}}y_{i,j}^{k,f}\right)=0,\forall i\in V,e_{ij}\in E$$
(12)$$x_{i,j}\left(\sum_{i\in V^{-}}y_{i,j}^{k,f}\right)av_{j}^{k}=0,\forall i,j\in V,e_{ij}\in E$$
(13)$$\sum_{k\in D}\left(av^{t,k}\right)\leq d$$
(14)$$t_{j}\geq t_{i}+w_{ij}/v^{Veh}-M\left(1-x_{ij}\right),\forall i,j\in V,i\neq j$$
(15)$$t_{j}^{k}\geq t_{i}^{k}+w_{ij}/v^{D}-M\left(1-y_{ij}^{k,f}\right),\forall i,j\in V,i\neq j$$
(16)$$t_{r}^{k}-t_{l}^{k}\leq P$$
(17)$$t_{r}-t_{l}^{k}\leq P$$
(18)$$z_{l,r}^{k,f}=1$$

Constraint (2) shows that for any node k, the number of inward and outward arcs are the same, which means that the number of edges the vehicle enters node k and exits from node k are the same. Constraints (3) and (4) ensure that the vehicle starts from the depot only once, and eventually returns to the depot. Constraints (5) and (6) ensure that the two types of target edges are accessed at least once. Constraints (7) and (8) ensure that each drone flight has only one launch node and one recovery node. Constraints (9) and (10) indicate that the coordination relationship between the drones and the vehicle is associated with the nodes in the vehicle that launch and recover the drones, ensuring that the launch and recovery nodes of drones must be nodes in the vehicle route.

There are also strict restrictions on the order in which drones can be launched and recovered. A drone can just be launched only if it has never been launched before, or it has been recovered since the last flight. If the drone has not been recovered at node i, it cannot be released from the vehicle. Constraints (11) and (12) stipulate that when the auxiliary variable $av_{i}^{k}=1$, the k-th drone cannot be launched or recovered at node i. In constraint (13), the number of drones flying at any time is limited to d. The time constraints of the vehicle and drones are shown in constraints (14) and (15). Constraints (14) and (15) represent the earliest arrival time of the vehicle and drones at each node, respectively, where M is an extremely large positive number. Constraints (16) and (17), respectively, express the constraints of the maximum patrolling time when the vehicle and the drones arrive at node r first, where l and r meet constraint (18); that is, the launch and recovery nodes of the k-th drone in the f-th flight are l and r, respectively.

## 4. Patrol Route Planning Approach

This section introduces the solution method for VD-ARP, namely the improved, adaptive, large neighborhood search algorithm (IALNS). It is mainly introduced from the following four parts: target edge encoding, initial solution generation, framework of IALNS algorithm, and neighborhood search strategies. All operations must satisfy the constraints described in Section 3.

### 4.1. Target Edge Encoding

Due to the limited battery capacity of drones, only a limited number of target edges can be accessed during a single flight. Consequently, we store the successively accessed target edges in the i-th drone flight in the set $SVTE_{i}$. Assuming that the first drone flight is shown in Figure 2, the drone is launched from node 2, and recovered at node 8 after visiting target edges $e_{53}$ and $e_{64}$, then $SVTE_{1}=\left\{e_{53},e_{64}\right\}$. The first drone route can be recorded as $〈2,SVTE_{1},8〉$.

The j-th pair of launch and recovery nodes for the i-th target edge is represented by $\left[l_{i,j},r_{i,j}\right]$, where l is the launch node and r is the recovery node. Each target edge assigned to the drone has a set of launch and recovery node pairs $LR_{i}=\left\{\left[l_{i,1},r_{i,1}\right],\left[l_{i,2},r_{i,2}\right],\ldots \right\}$ that satisfy the drone battery capacity constraint, where each pair of nodes l and r should meet constraint (16–18) in Section 3. The target edges accessed by a drone flight have the same set of launch and recovery node pairs.

The variable A, used to store the target edge information, is a matrix variable of size N×6, where N is the number of target edges. Each row of this matrix stores one target edge information. The first column stores visitors of the target edges, where -1 means accessed by the vehicle, and 1 by drones. The number i in the second column means that the target edge belongs to the set $SVTE_{i}$. The third column represents the order in which the target edges are accessed, labeled 1 to N. The access order of target edges in the same set is contiguous. The fourth column stores the access direction of the target edges, wherein 0 indicates the forward direction that the target edge $e_{ij}$ is accessed from i to j, and 1 indicates the reverse direction. The fifth and sixth columns store the start and end nodes accessing the target edges. If the target edge is accessed by the vehicle (marked -1 in the first column), the start node and the end node are the two endpoints of the target edge, respectively. If it is accessed by a drone (marked as 1 in the first column), then a pair of launch and recovery nodes are stored, and the target edges in the same set are marked with the same node pairs. As shown in Figure 3, target edge 1 and target edge 3 belong to the same set, that is, they are accessed by the same drone flight. Therefore, the start nodes and end nodes of these two target edges are the same.

### 4.2. Initial Solution Generation

First, the target edge information in the matrix variable A, and the launch and recovery node pairs $LR_{i}=\left\{\left[l_{i,1},r_{i,1}\right],\left[l_{i,2},r_{i,2}\right],\ldots \right\}$ of the target edges assigned to drones, are randomly generated. Then, Algorithm 1, which gives the “Drone-First, Vehicle-Then” heuristic method, is designed to generate the initial vehicle and drone routes.

The heuristic method first initializes the vehicle route, drone route, and the number of drone routes f (line 1), and obtains the set of vehicle target edges TV and the set of drone target edges TD. It is worth noting that TV and TD here are the sets of target edges that have been determined and assigned to the vehicle and the drones, respectively. In the main loop of generating the initial solution (lines 4–13), the drone routes $\sum_{f}S^{dr,f}$ are generated first (lines 4–8), and then the vehicle route $S^{veh}$ is generated, until all target edges are assigned.

The process of generating a drone route $S^{dr,f}$ can be described as: for the target edge i assigned to the drones, randomly select a pair of launch and recovery nodes in $LR_{i}$, and then determine the drone route according to the direction and access order of all target edges in the same set as i. Finally, delete the target edges from TD. Except for accessing the target edges, the drones fly along the shortest distance between any two nodes in the road network, and the distance between the two nodes passed by a drone is the Euclidean distance [46].

The process of generating the vehicle route $S^{veh}$ can be described as follows: first, the launch and recovery nodes of each drone route are continuously inserted into the vehicle route, according to the access order. Next, for the target edge i assigned to the vehicle, two endpoints are inserted into adjacent positions in the vehicle route, according to the direction and access order. Finally, the target edge i is removed from the set TV. The vehicle can only travel along the road network, so the shortest distance between any two nodes in the vehicle route is calculated by Dijkstra algorithm [47].
**Algorithm 1****:** “Drone-First, Vehicle-Then” heuristic method
**Input:** The target edge information A; $\mathrm{launch}\;\mathrm{and}\;\mathrm{recovery}\;\mathrm{node}\;\mathrm{pairs}\;LR_{i}$
**Output:** Initial solution *S*1**Initialize** vehicle route $S^{veh}\leftarrow []$; the f-th drone route $S^{dr,f}\leftarrow []$; f←02**GET** TV and TD from A
3**While**TV⋃TD≠∅ **do**4  **If**
i∈TD **then**5     
f←f+1
6     Randomly select a pair of launch and recovery nodes, and determine a     drone route $S^{dr,f}$ according to the direction and access order of each     target edge in the same set as i
7     Delete all target edges that are in the same set as target edge i from TD
8     Insert the launch and recovery nodes of target edge i into the vehicle route $S^{veh}$     according to the access order9  **Else if**
i∈TV **then**10     Insert the two endpoints of target edge i into the vehicle route $S^{veh}$     according to the access order11     Delete target edge i from TV
12  
**End if**
13**End while**14$S=S^{veh}⋃\sum_{f}S^{dr,f}$

### 4.3. Framework of IALNS Algorithm

IALNS takes a large neighborhood search [48] as the basic search framework, and integrates a variety of effective mechanisms, such as adaptive neighborhood selection mechanism, tabu table, simulated annealing mechanism, initial solution generation method, and neighborhood search strategies, to improve the performance of IALNS. The objective of IALNS is to minimize the total patrol time. Algorithm 2 describes the framework of IALNS in detail.

IALNS inputs the target edge information A, initial temperature $T_{0}$, final temperature $T_{min}$, maximum number of iterations $I_{max}$, annealing rate γ, destroy operator set DO, and repair operator set RO, and outputs the optimal solution $S_{best}$. First, the initial solution S is obtained by the “Drone-First, Vehicle-Then” heuristic method introduced in Section 4.2. Then, the weight of the destroy operators do, weight of the repair operators ro, as well as current iteration I, current temperature T, and optimal solution $S_{best}$ are initialized in line 2. In the main loop of IALNS, the solution is iteratively updated until the termination criterion is satisfied (lines 3–19). The destroy operator and the repair operator are selected and executed to obtain a new solution $S^{\prime }$ (line 4–5), and the new solution is accepted according to the simulated annealing mechanism. If the optimal solution is worse than the new solution, accept the new solution $S^{\prime }$; otherwise, accept $S^{\prime }$ with a probability exp(△f/T)>ζ, where ζ is randomly sampled in the interval [0,1) (line 7–14). Finally, at the end of each main loop, update do and ro, the tabu table, the current iteration I, and current temperature T accordingly (lines 15–18).
**Algorithm 2:** Framework of IALNS
**Input:**A; $T_{0}$; $T_{min}$; $I_{max}$; γ; DO; RO
**Output:** $S_{best}$1$S\leftarrow DFVT\left(A,SE_{i}\right)$2**Initialize**do=(1,…,1); ro=(1,…,1); I←1; $T\leftarrow T_{0}$; $S_{best}\leftarrow S$3**While** $T>T_{min}$ and $I<I_{max}$ **do**4  Select the destroy operator d∈DO and repair operator r∈RO according  to do and ro
5  
$S^{\prime }=r\left(d\left(S\right)\right)$
6  
$△f=F\left(S^{\prime }\right)-F\left(S\right)$
7  **If** accept $S^{\prime }$ **then**8    
$S\leftarrow S^{\prime }$
9    **If** $F\left(S\right)<F\left(S_{best}\right)$ **then**10      
$S_{best}\leftarrow S^{\prime }$
11    **Else if**
exp(△f/T)>ζ **then**12      
$S_{best}\leftarrow S^{\prime }$
13    
**End if**
14  
**End if**
15  Update do and ro
16  Update the tabu table 17  
I←I+1
18  
T←γT
19**End while**

### 4.4. Neighborhood Search Strategies

The neighborhood search strategies designed in this paper are composed of destroy operators and repair operators in pairs. The destroy operators delete the target edges from the original vehicle route and perform certain operations on the target edges, while the repair operators reinsert the operated target edges into the vehicle route with a certain rule to update the vehicle route.

Since the encoding of target edges in the matrix variable A is randomly generated, changing these elements may improve the solution in the neighborhood search. According to the characteristics of the proposed DV-ARP, we design a total of 7 destroy operators, which are described as follows:

#### 4.4.1. Change the Access Direction of Target Edges Assigned to Drones

Randomly select a target edge assigned to drones and change the access direction of target edges, as Figure 4 shows. If there is only one target edge in the drone route, the access direction of the target edge is changed, and the start and end nodes are exchanged, and then deleted from the vehicle route. If there are multiple target edges, the entire drone route is flipped, and the start and end nodes are deleted from the vehicle route.

#### 4.4.2. Change the Access Direction of a Target Edge Assigned to the Vehicle

Select a target edge accessed by the vehicle randomly, change the access direction, and delete the endpoints of the target edge from the vehicle route, as shown in Figure 5.

#### 4.4.3. Change the Start and End Nodes of a Drone Route

A target edge accessed by drones is randomly selected and changed, as Figure 6 displays. Randomly re-select a pair of launch and recovery nodes in $LR_{i}$, and delete the original launch and recovery nodes from the vehicle route.

#### 4.4.4. Delete Target Edge

Randomly select a target edge, if it is accessed by a drone, delete the start and end nodes from the vehicle route, as in Figure 7a. If it is accessed by the vehicle, the endpoints of the target edge are directly deleted from the vehicle route, as in Figure 7b.

#### 4.4.5. Merge Two Drone Routes

Randomly choose two drone routes, and merge the target edges in the two routes to form one drone route. Then, the original two pairs of start and end nodes are deleted from the vehicle route, which is depicted in Figure 8.

#### 4.4.6. Reorganize Two Drone Routes

Randomly choose two drone routes and select a target edge from each route separately. Put these two target edges in one set, and put the remaining target edges of the two drone routes in another set. The start and end nodes of the two sets are randomly generated, and the two drone routes are formed accordingly. Delete the original start and end nodes from the vehicle route. The operator is presented in Figure 9.

#### 4.4.7. Change the Assignment Object of the Target Edge

First, randomly select a target edge. If it is accessed by a drone, determine whether the target edge can be assigned to a vehicle. If so, assign the target edge to the vehicle, and delete the start and end nodes from the vehicle, as shown in Figure 10a. If it is accessed by the vehicle, randomly generate a pair of start and end nodes of the target edge to form a new drone route, and the two endpoints are deleted from the vehicle route. In Figure 10b, the target edge $e_{78}$ originally assigned to the vehicle is reassigned to the drones.

#### 4.4.8. The Repair Operator

The repair operator can be abstracted, as in Figure 11. The two endpoints of the target edge accessed by the vehicle, or the start and end nodes of the target edge accessed by the drone after the destroy operator operation, are randomly inserted into the vehicle route. Note that the two endpoints of the target edge assigned to the vehicle must be inserted into adjacent positions.

In any destroy operator operations, if the number of drone flights increases, it is necessary to determine whether the number of drones performing tasks during the interval of a newly added drone flight is less than the number of drones carried by the vehicle. If not, skip this operation to ensure that the constraints (11–13) in Section 3 are satisfied.

The combination of the above seven destroy operators and the repair operator constitutes seven different neighborhood search strategies.

## 5. Experiments and Discussion

In this section, we first set up the experimental parameters, designed the instances of different scales, and introduced the comparison algorithms. Next, we tested the performance of the proposed algorithm through simulation experiments. Finally, a practical case was applied, to further verify the applicability of the algorithm. All algorithms were implemented in Python, and run on an x64 Windows 10 computer, with I7-10700F CPU and 16.00 GB memory.

### 5.1. Simulation Setup

According to previous experiments, the parameters involved were configured as follows: for each instance, the maximum number of iterations $I_{max}=1500$. The initial temperature $T_{0}$ was set to 100, and the termination temperature $T_{min}=0.1$. The annealing rate γ is 0.99. In addition, the relevant parameters of the vehicle and drones were set according to typical situations in practical applications, as shown in Table 3.

Considering the characteristics of the urban road network, an undirected network graph with 50 nodes and 79 edges was randomly generated within the range of 5 km × 5 km. Three groups of instances with 5, 10, and 20 target edges were designed, marked as E1/E2/E3, and different target edges (distinguished by A, B, and C) were randomly set for each group of instances. For each instance, the number of target edges that can only be accessed by the drones is 1/5 of the total number of target edges. Table 4 describes the instances in detail.

In this paper, the proposed IALNS was compared with VND [49], VND with tabu list (VND-tabu), and the improved LNS [48] (ILNS). Compared with IALNS, the neighborhood search strategies were randomly selected with no weight change in ILNS. Moreover, the length of the table list is set as follows:(19)$$L_{max}=2\left|\sqrt{\frac{N\left(N-1\right)}{2}}\right|\;$$
where N represents the total number of target edges.

### 5.2. Experiments and Analyses

All comparison algorithms used the same neighborhood search strategies and initial solution generation method as IALNS, for fairness. We ran each algorithm 20 times for each instance. Table 5 and Figure 12 present the average performance of IALNS and the three comparison algorithms. “Result (s)” and “Running time (s)” represent the total patrol time and consumed running time, respectively. “Average” is the average value of the nine instances, and “GAP” is a parameter used to further compare the performance differences between IALNS and the other approaches, which can be calculated by
(20)$$\mathrm{GAP}\left(X,\mathrm{IALNS}\right)=\frac{\left(f_{X}-f_{\mathrm{IALNS}}\right)}{\max\left(f_{X},f_{\mathrm{IALNS}}\right)}\times 100\%\;$$

In Table 5 and Figure 12, it can be observed that as the number of target edges increases, the solution results and running time of each algorithm increase. From Table 5, the comparison of all algorithms shows the superiority of IALNS. Compared with the three comparison algorithms, IALNS reduces the total patrol time by 6.56%, 7.45%, and 3.62% respectively. As Figure 12 displays, ILNS and IALNS always perform better solutions. It is because ILNS and IALNS have a larger search space, by randomly selecting the neighborhood search strategies, which increases the possibility of finding better solutions. Compared with ILNS, IALNS has obvious advantages in the solution results, which can be explained as the adaptive selection mechanism dynamically adjusting the weight of each neighborhood search strategy to guide the search direction, in order to obtain better solutions. Moreover, IALNS performs best in terms of solution quality and running speed. In general, the IALNS algorithm combines the advantages of the adaptive mechanism, simulated annealing mechanism, and tabu table, which not only avoids falling into local optimum, but also improves the convergence speed.

Taking the three instances E1-A, E2-A, and E3-A as representatives, the schematic diagrams of the route planning results are shown in Figure 13. The vehicle routes are represented by solid black lines, and the routes of different drones are labeled in different colors. The forward directions of all routes are indicated by arrows. The blue dots label the depot. Target edges that can be accessed by the vehicle or drones are represented by red segments, and target edges that can only be accessed by drones are represented by pink segments.

To further verify the stability of each algorithm, the statistical results of the standard deviation of different algorithms are obtained, as shown in Table 6. Each algorithm was independently run 20 times. “Max”, “Min” and “Avg” are the maximum, minimum, and average values of the standard deviations of the nine instances, respectively.

The statistical results in Table 6 show that the average standard deviation of IALNS is 0.0826, which is smaller than that of the other comparison algorithms. The standard deviation of the VND-tabu algorithm performs better than that of VND, due to the tabu table, which reduces the repeated search of the same target edge in a short term. The standard deviation of IALNS is smaller than that of ILNS, indicating that the adaptive selection mechanism of the neighborhood search strategies improves the stability of the algorithm, to a certain extent.

In order to verify the impact of the number of drones on the experimental results, a sensitivity analysis based on the three instances, E1_A, E2_A, and E3_A is designed. For each instance, the number of drones is set from one to nine. Each algorithm is run 10 times repeatedly. All experimental results are recorded in Figure 14 and Table 7.

As depicted in Figure 14 and Table 7, in the three instances, as the number of drones increases, the experimental results gradually decrease, and the IALNS algorithm still has obvious advantages. In particular, the reduction is greatest when the number of drones increases from one to three. However, when the number of drones increases from three to nine, the improvement effect on the experimental results decreases with each additional drone. When the number of drones increases by a certain value, the drone resources are saturated, and the drones are enough to meet the patrol needs. At this time, even if the number of drones carried by the vehicle increases, the increased drones will not be dispatched for patrolling. As Figure 14 displays, for instances E1_A and E2_A, three drones can saturate, while E3_A may need six drones (although the gain from increasing the drones from three to six is small). The number of drones currently taken matches the actual demand for traffic patrols. Assigning more drones to the vehicle might save patrol time, but the improvement in cost is huge. Therefore, the choice of the number of drones shows a trade-off between patrol time and cost.

When the approach proposed in this paper is applied to the smart city context, the speed of the vehicle may be significantly improved, and the experimental results may vary greatly. Therefore, we also take the instances E1_A, E2_A, and E3_A as examples to design analysis experiments on the speed of the vehicle. The vehicle speed is set at 30–60 km/h, with an interval of 5 km/h. Each algorithm is run 10 times independently. Figure 15 and Table 8 show the experimental results at different speeds of the vehicle.

The effects of the speed of drones on the performances of the algorithms are obvious. As Figure 15 and Table 8 display, in the three instances, the experimental results decrease significantly as the vehicle speed increases, and IALNS always performs best. It follows that the increased speed of the vehicle is an effective consideration to reduce the patrol time, which puts forward higher requirements for the state of the urban traffic network. However, when the vehicle speed is doubled from 30 km/h to 60 km/h, the total patrol time is reduced by less than half. This is because, despite the increase in vehicle speed, the unchanging speed of drones, and the potential increase in waiting time, make the reduction in patrol time less than desirable.

### 5.3. An Actual Case Study

In order to further study the application significance of the VD-ARP proposed in this paper, an actual case in the real world is considered. The simplified road network in a certain area of Changsha City is shown in Figure 16. There are 56 intersections and 94 road sections in this area, which are represented by small circles and line segments, respectively. The eight road sections that can be patrolled by vehicles or drones, and two road sections that can only be patrolled by drones are randomly set, marked with red and pink lines, respectively.

The parameter settings are the same as in Section 5.1. The patrol time (second) with different numbers of drones is presented in Table 9, and the route planning of the vehicle and the drones, obtained by IALNS, is shown in Figure 17.

In this case study, by setting appropriate parameters, such as the vehicle speed, drone speed, and the number of drones, a feasible route planning scheme for the vehicle and drones is obtained within about 40 s of running time, which meets the needs of urban traffic patrols. There are 10 road sections to be patrolled, with a total length of 14.56 km. The vehicle routes are represented by solid black lines, and the routes of different drones are labeled in different colors. It can be seen from Table 9 that when the number of drones is three, the patrol time obtained by the IALNS algorithm is 2283 s, that is, 0.63 h. Hence, a satisfactory solution is obtained within an acceptable time. Compared with the other three comparison algorithms, IALNS has obvious advantages. It is worth noting that in the route planning scheme given in Figure 17, there are at most two drones patrolling at the same time, that is, at least two drones are needed. The additional drones are barely effective. Through the coordination of multiple drones and a vehicle, the efficiency of urban traffic patrols is significantly improved.

## 6. Conclusions

Proper traffic patrols improve the safety of urban traffic. To improve the efficiency and flexibility of traffic patrol, a coordinated vehicle–drone arc routing problem, with a vehicle and multiple drones (DV-ARP), is proposed for planning the parallel routes of a vehicle and drones in urban traffic patrols. To solve the problem, an improved, adaptive, large neighborhood search algorithm (IALNS) is designed. Firstly, the initial solution is generated by the heuristic method of “Drone-First, Vehicle-Then”. Secondly, a set of destroy operators and a repair operator are designed to generate new candidate solutions. Finally, the simulated annealing mechanism, tabu table, and multiple neighborhood search strategies are integrated into the framework of an adaptive large neighborhood search algorithm, in order to search the solution space and find the optimal solution.

Experiments with different dimensions are conducted to explore the effectiveness of IALNS. The superiority of IALNS is verified by numerical experiments compared with other algorithms on different scales. The results show that the proposed approach IALNS reduces the total patrol time by at most 7.45%, compared with the three comparison algorithms (i.e., VND, VND-tabu, and ILNS). Additionally, the performance of all the four algorithms shows that all of them address the instances within different scales. To further test the impact of the main parameters on the results, two critical factors are evaluated, including the drone number carried by the vehicle and the vehicle speed. As the number of drones increases from one to three, the total patrol time has a dramatic decline, then slows down, and finally stops decreasing. It means that patrol scenarios with different numbers of target edges have different requirements for the drones. In addition, the total patrol time also decreases as the vehicle speed increases from 30 to 60 km/h. A real-world road network in Changsha City is simulated to further verify the applicability of IALNS.

The proposed IALNS algorithm in this paper is expected to provide effective route planning schemes for traffic management departments to carry out urban traffic patrols or collect traffic data. However, it ignores some complex constraints that may need to be considered in practical applications, such as energy consumption and separated launching/recovering time of drones, as well as dynamic demand, which would be further studied in the future. In addition, how to design a reasonable evaluation system to explore the balance between drone costs and patrol efficiency is also worth studying.

## Figures and Tables

**Figure 1 sensors-22-03702-f001:**
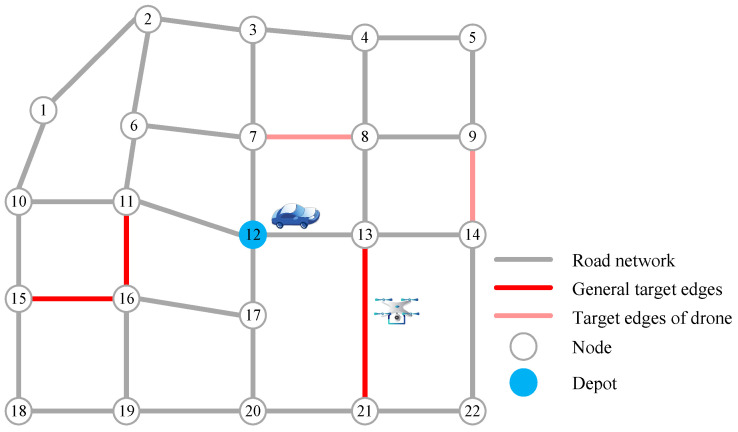
Schematic diagram of road network and target edges, where the numbers 1–22 represent different nodes of the road network.

**Figure 2 sensors-22-03702-f002:**
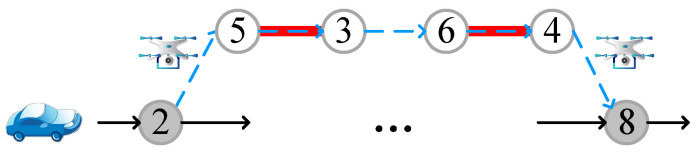
An example of a drone flight, where the numbers represent different nodes of the road network.

**Figure 3 sensors-22-03702-f003:**
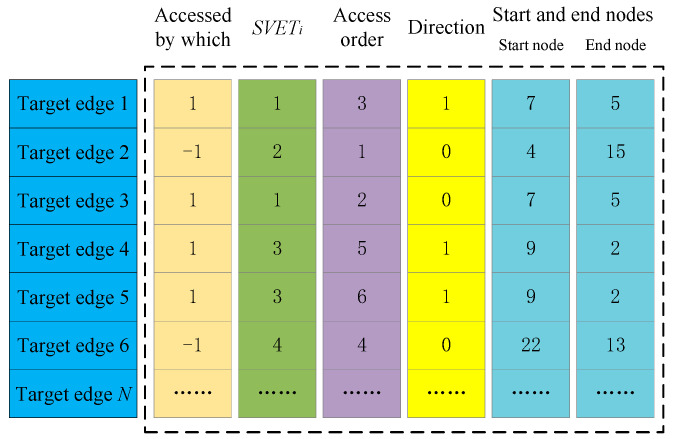
Encoding of matrix variable A.

**Figure 4 sensors-22-03702-f004:**
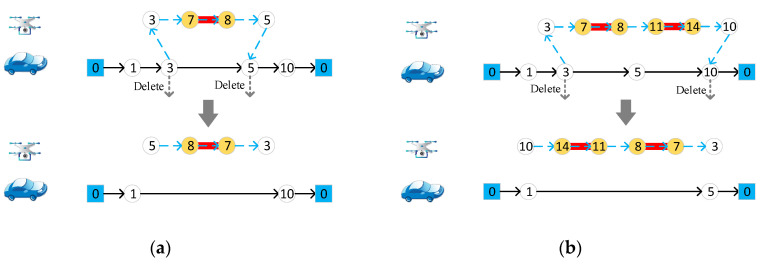
Change the access direction of target edges assigned to drones. (**a**) For one target edge in the drone route; (**b**) for multiple target edges. Different numbers in the figure represent different nodes of the road network.

**Figure 5 sensors-22-03702-f005:**
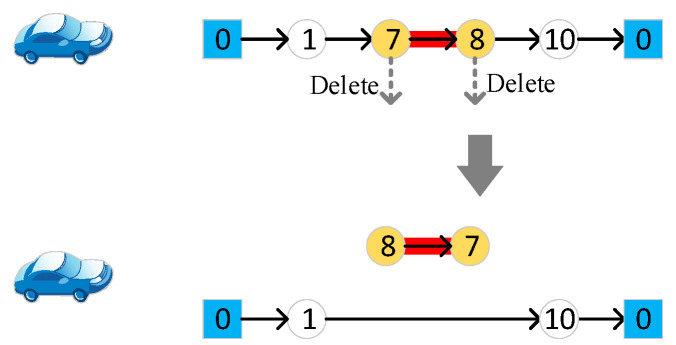
Change the access direction of a target edge assigned to the vehicle. Different numbers in the figure represent different nodes of the road network.

**Figure 6 sensors-22-03702-f006:**
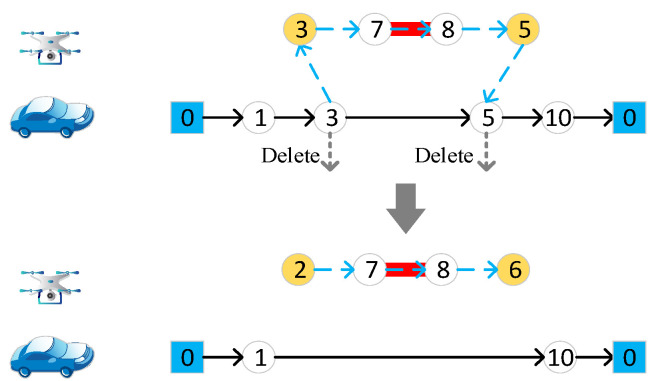
Change the start and end nodes of a drone route. Different numbers in the figure represent different nodes of the road network.

**Figure 7 sensors-22-03702-f007:**
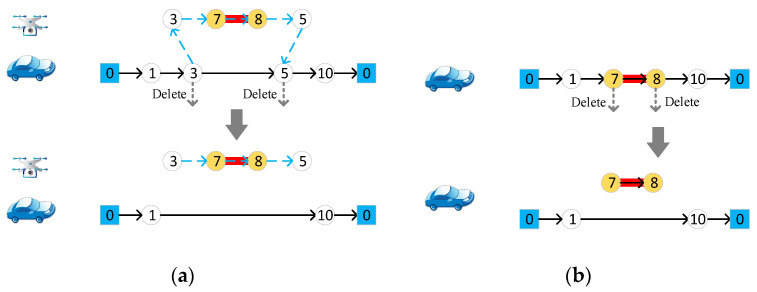
Delete target edge. (**a**) Delete target edge accessed to drones; (**b**) delete target edge accessed to vehicle. Different numbers in the figure represent different nodes of the road network.

**Figure 8 sensors-22-03702-f008:**
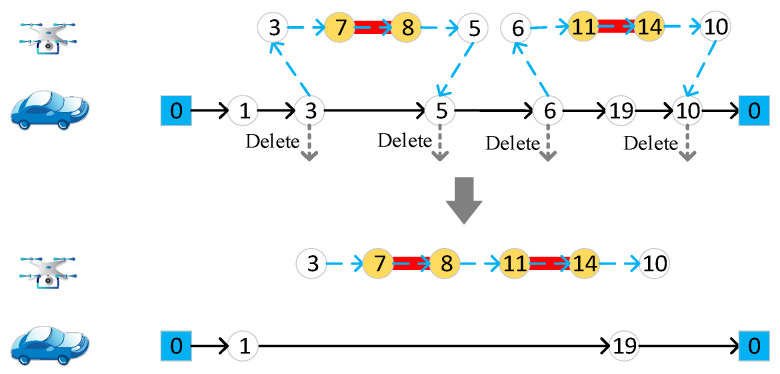
Merge two drone routes. Different numbers in the figure represent different nodes of the road network.

**Figure 9 sensors-22-03702-f009:**
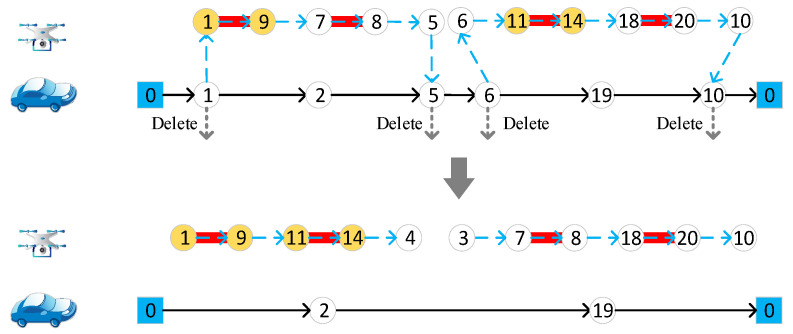
Reorganize two drone routes. Different numbers in the figure represent different nodes of the road network.

**Figure 10 sensors-22-03702-f010:**
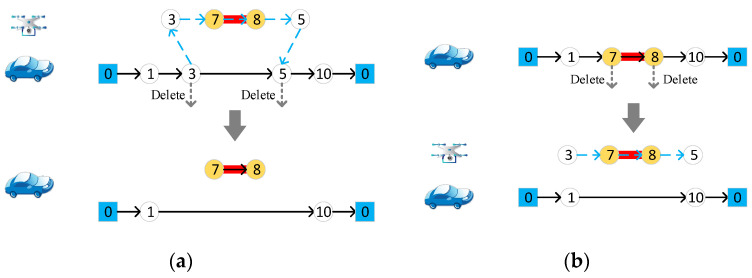
Change the assignment object of the target edge. (**a**) Drone to vehicle; (**b**) vehicle to drone. Different numbers in the figure represent different nodes of the road network.

**Figure 11 sensors-22-03702-f011:**
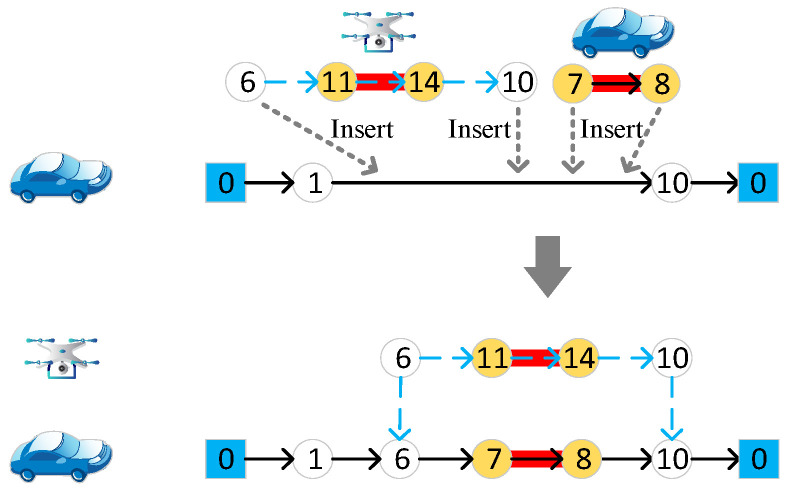
The repair operator. Different numbers in the figure represent different nodes of the road network.

**Figure 12 sensors-22-03702-f012:**
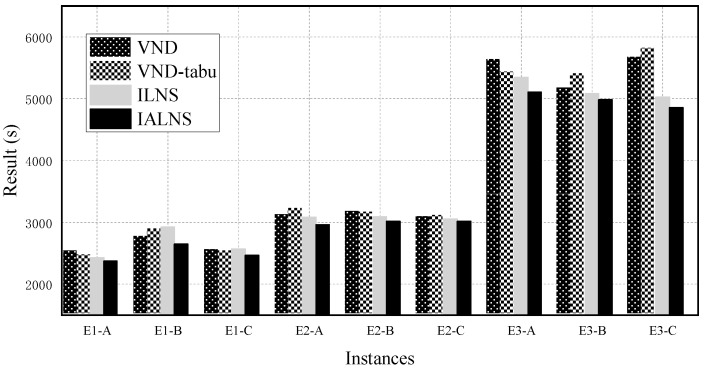
Comparison of results of different algorithms.

**Figure 13 sensors-22-03702-f013:**
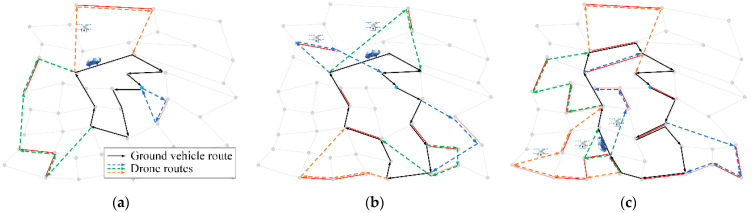
Schematic diagrams of route planning results on different instances with different number of target edges. (**a**) E1-A; (**b**) E2-A; (**c**) E3-A.

**Figure 14 sensors-22-03702-f014:**
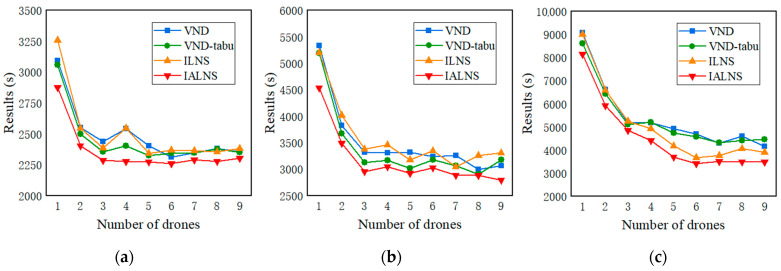
Impact of different number of drones on experimental results. (**a**)E1-A; (**b**) E2_A; (**c**) E3-A.

**Figure 15 sensors-22-03702-f015:**
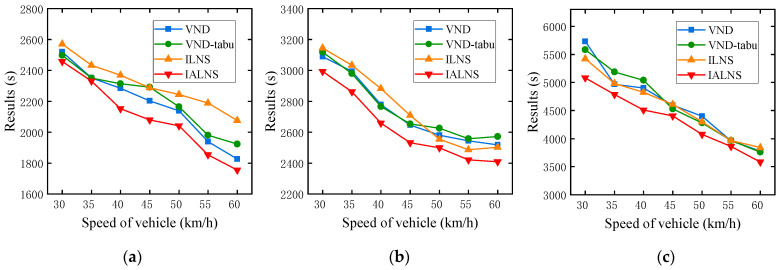
Impact of different speeds of the vehicle on experimental results. (**a**)E1-A; (**b**) E2_A; (**c**) E3-A.

**Figure 16 sensors-22-03702-f016:**
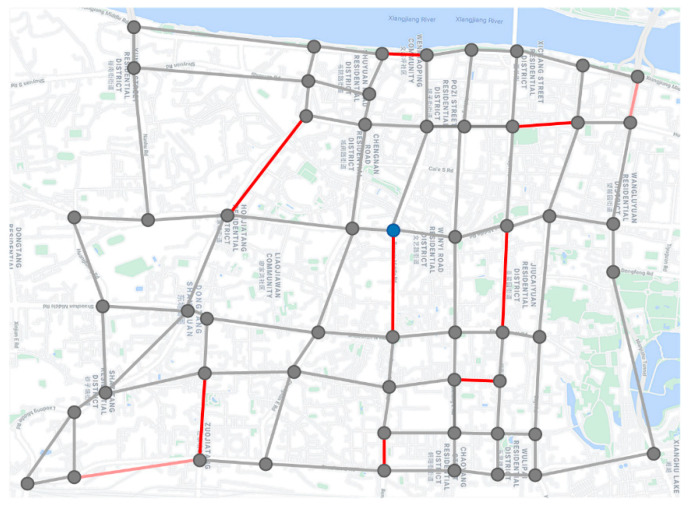
Simplified road network in a certain area of Changsha City.

**Figure 17 sensors-22-03702-f017:**
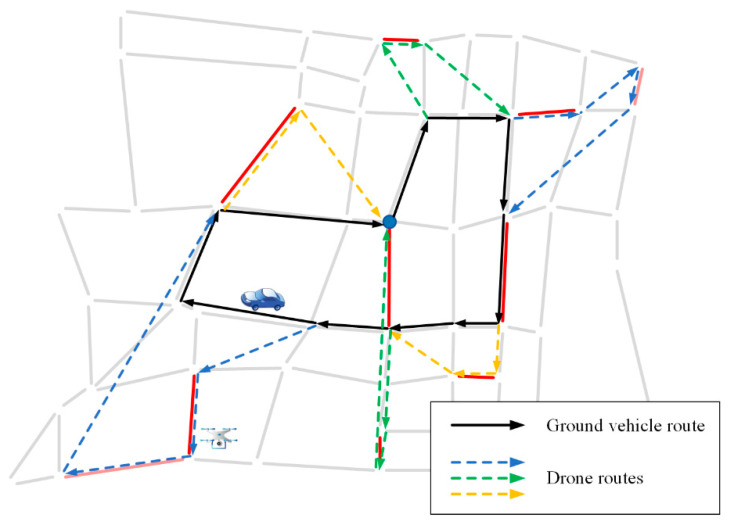
Route planning of the vehicle and the drones.

**Table 1 sensors-22-03702-t001:** Summary of related works.

References	Research Topic	Vehicles	Drones	Objective	Task Type	Network
Jalili et al. [23]	Traffic safety management	M	-	-	-	Yes
Li et al. [24]	Evaluation system	M	-	-	-	Yes
Wang et al. [26]	Data analysis	M	n	-	-	Yes
Kwan [37]	Routing optimization	1	-	Cost	Arc	Yes
Hertz et al. [39]	Routing optimization	1	-	Cost	Arc	Yes
Babaee et al. [42]	Routing optimization	M	-	Cost	Arc	Yes
Campbell et al. [44]	Routing optimization	-	1	Cost	Arc	Yes
Murray et al. [4]	Routing optimization	1	1	Time	Point	No
Dorling et al. [3]	Routing optimization	M	n	Time/Cost	Point	No
Agatz et al. [6]	Routing optimization	1	1	Time	Point	Yes
Hu et al. [10]	Routing optimization	1	n	Time	Point	Yes
Luo et al. [20]	Routing optimization	1	1	Time	Point & Arc	Yes
This Work	Routing optimization	1	n	Time	Arc	Yes

**Table 2 sensors-22-03702-t002:** Symbols and descriptions.

Parameters	Descriptions
D	Set of drones, D={1,2,…,d}
A	Matrix variable to store information of target edges
$SVTE_{i}$	Set of successively visited target edges in the *i*-th drone flight
$\left[l_{i,j},r_{i,j}\right]$	The *j*-th pair of launch and recovery nodes for the *i*-th target edge
R	$\mathrm{Vehicle}\;\mathrm{route},\;R=\left\{v_{0},\ldots ,v_{PC}\right\}$
$l,SVTE_{i},r$	$\mathrm{The}\;\mathrm{drone}\;\mathrm{route}\;\mathrm{which}\;\mathrm{visits}\;\mathrm{the}\;\mathrm{target}\;\mathrm{edges}\;\mathrm{in}\;\mathrm{set}\;SVTE_{i}$, with *l* for the launch node and *r* for the recovery node
$w_{ij}$	$\mathrm{Length}\;\mathrm{of}\;\mathrm{edge}\;e_{ij}$
$t_{i}$	The time when the vehicle arrives at node *i*
$t_{i}^{k}$	The time when the *k*-th drone arrives at node *i*
$\varepsilon_{i}$	The time interval required for the vehicle to wait for the recovery of the drones after arriving at the node *i*
$v^{Veh}$	Vehicle speed
$v^{D}$	Drone speed
P	Maximum flight time of a drone
**Decision variables**	**Descriptions**
$x_{ij}$	$x_{ij}=1$ $\;\mathrm{if}\;\mathrm{the}\;\mathrm{vehicle}\;\mathrm{accesses}\;\mathrm{the}\;\mathrm{edge}\;e_{ij}$ $,\;\mathrm{otherwise},\;x_{ij}=0$
$y_{i,j}^{k,f}$	$y_{i,j}^{k,f}=1$ if the *f*-th flight of the *k*-th drone accesses the edge *e_ij_*, otherwise, $y_{i,j}^{k,f}=0$
$z_{i,j}^{k,f}$	$z_{i,j}^{k,f}=1$ if the *f*-th flight of the *k*-th drone is launched from *i* and recovered at *j*, otherwise, $z_{i,j}^{k,f}=0$
$av_{i}^{k}$	$\mathrm{Auxiliary}\;\mathrm{variable},\;av_{i}^{k}=0$ if the *k*-th drone is launched or recovered at *i*, otherwise, $av_{i}^{k}=1$
$av^{t,k}$	$\mathrm{Auxiliary}\;\mathrm{variable},\;av^{t,k}=1$ if the *k*-th drone has been launched at time *t*, otherwise, $av^{t,k}=0$

**Table 3 sensors-22-03702-t003:** Experimental parameter design.

Parameters	Value (Unit)
Number of drones	3
Vehicle speed	30 km/h
Drone speed	35 km/h
Battery life of a drone	0.67 h

**Table 4 sensors-22-03702-t004:** Introduction of the instances.

No.	Instances	Total Num. of Target Edges	Num. of Target Edges Only for Drones
1	E1-A	5	1
2	E1-B	5	1
3	E1-C	5	1
4	E2-A	10	2
5	E2-B	10	2
6	E2-C	10	2
7	E3-A	20	4
8	E3-B	20	4
9	E3-C	20	4

**Table 5 sensors-22-03702-t005:** The results and running time of each instance on IALNS and the other three comparison algorithms.

Instances	VND		VND-Tabu		ILNS		IALNS	
	Result (s)	Running Time (s)	Result (s)	Running Time (s)	Result (s)	Running Time (s)	Result (s)	Running Time (s)
E1-A	2521	27.85	2452	27.42	2430	23.96	2376	27.92
E1-B	2751	21.18	2878	25.56	2923	11.42	2647	22.18
E1-C	2542	25.02	2521	25.93	2570	21.29	2466	27.32
E2-A	3108	40.16	3212	43.25	3086	37.56	2960	36.30
E2-B	3163	46.78	3153	47.77	3092	43.49	3013	43.78
E2-C	3079	38.20	3098	40.86	3054	37.47	3015	36.18
E3-A	5630	106.58	5432	104.03	5345	92.92	5108	90.36
E3-B	5167	120.09	5398	127.58	5081	105.68	4986	105.48
E3-C	5671	90.25	5817	96.22	5027	82.86	4855	73.58
Average	3737	-	3773	-	3623	-	3492	-
GAP (%)	6.56%	-	7.45%	-	3.62%	-	0	-

**Table 6 sensors-22-03702-t006:** The statistical results of the standard deviation of different algorithms.

Instances	Total Num. of Target Edges	Algorithms			
		VND	VND-Tabu	ILNS	IALNS
E1-A	5	0.0829	0.0777	0.0681	0.0770
E1-B	5	0.1341	0.0496	0.0519	0.1312
E1-C	5	0.0660	0.0551	0.0638	0.0580
E2-A	10	0.1399	0.1620	0.0953	0.0710
E2-B	10	0.1195	0.0560	0.0511	0.0514
E2-C	10	0.0690	0.0918	0.0542	0.0849
E3-A	20	0.1329	0.1663	0.0902	0.1262
E3-B	20	0.1938	0.1945	0.2235	0.0792
E3-C	20	0.2432	0.2603	0.2003	0.0643
Max		0.2432	0.2603	0.2235	0.1312
Min		0.0660	0.0496	0.0511	0.0514
Avg		0.1313	0.1237	0.0998	0.0826

**Table 7 sensors-22-03702-t007:** Experimental results of each algorithm under different number of drones.

Instances	Algorithms	Num. of Drones								
		1	2	3	4	5	6	7	8	9
E1-A	VND	3095	2551	2438	2542	2403	2313	2345	2383	2357
	VND-tabu	3057	2498	2353	2404	2326	2344	2347	2378	2350
	ILNS	3256	2547	2386	2544	2341	2366	2363	2356	2379
	IALNS	2876	2403	2287	2278	2274	2260	2288	2276	2303
E2-A	VND	5332	3822	3319	3307	3318	3234	3256	2989	3065
	VND-tabu	5194	3671	3121	3166	3011	3172	3061	2905	3173
	ILNS	5205	4017	3374	3460	3179	3351	3049	3259	3307
	IALNS	4538	3494	2952	3048	2920	3025	2885	2878	2792
E3-A	VND	9074	6612	5186	5187	4912	4684	4306	4596	4152
	VND-tabu	8599	6427	5114	5212	4731	4580	4328	4405	4471
	ILNS	9003	6600	5253	4917	4178	3662	3774	4062	3907
	IALNS	8146	5920	4853	4413	3688	3413	3518	3499	3480

**Table 8 sensors-22-03702-t008:** Experimental results of each algorithm under different speeds of the vehicle.

Instances	Algorithms	Speed of the Vehicle						
		30	35	40	45	50	55	60
E1-A	VND	2520	2352	2284	2203	2138	1938	1826
	VND-tabu	2497	2351	2314	2292	2165	1981	1923
	ILNS	2571	2434	2370	2287	2244	2188	2075
	IALNS	2458	2331	2153	2080	2041	1914	1754
E2-A	VND	3090	2993	2779	2647	2580	2543	2417
	VND-tabu	3119	2980	2767	2654	2626	2557	2572
	ILNS	3145	3033	2882	2708	2554	2486	2503
	IALNS	2993	2862	2658	2531	2498	2421	2409
E3-A	VND	5733	4967	4902	4596	4401	3960	3778
	VND-tabu	5585	5187	5039	4527	4279	3965	3759
	ILNS	5423	4981	4826	4608	4297	3963	4042
	IALNS	4853	4786	4508	4404	4076	3861	3582

**Table 9 sensors-22-03702-t009:** Patrol time (s) with different numbers of drones.

Algorithms	Num. of Drones					
	1	2	3	4	5	6
VND	3352	2777	2386	2292	2384	2441
VND-tabu	3268	2516	2353	2202	2198	2347
ILNS	3146	2391	2371	2211	2078	2294
IALNS	2980	2265	2283	2109	2002	2184

## Data Availability

Not applicable.

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
