# Peer review of "A Coordinated Vehicle–Drone Arc Routing Approach Based on Improved Adaptive Large Neighborhood Search"

_sensors, 2022, doi:10.3390/s22103702_

Round 1

Reviewer 1 Report

This paper studies a vehicle drone arc routing problem (VD-ARP) consisting of one vehicle and multiple drones. To solve this problem, a mathematical model with minimum total patrol time is constructed and an improved adaptive large neighborhood search algorithm (IALNS) is proposed. This work is interesting. However, the author should revise the manuscript carefully. Especially, pay more attention to the following several cases.

1. Some contents in the paper are not described very clearly and need to be supplemented.

(1) The relationship between r and l in constraints (16) and (17) of the mathematical model is not clear.

(2) There is no inevitable connection between the mathematical model in Section 3 and the IALNS algorithm in Section 4. The author needs to explain the role of this model in the design of IALNS algorithm in the manuscript.

(3) Line 7 of Algorithm 1 is not clearly described. What is the meaning of ‘all target edges in the set of target edge i’?

(4) The description of Figure 16 is not very clear. It should be indicated that the route planning result is in the case of three drones.

2. It can be seen from Figure 14, Table 6 and Table 7 that with the increase of the number of drones, the total patrol time obtained by the algorithm does not necessarily decrease. What is the reason? It needs careful analysis.

3. There are several obvious mistakes in the paper. For example:

(1) The last two sentences of Section 1 need to be combined and rewritten.

(2) Constraint (3) in the mathematical model is written incorrectly. It is exactly the same as constraint (2) now.

(3) Constraint (4) in the mathematical model is written incorrectly, which can’t ensure that the vehicle eventually returns to the depot.

(4) In Constraints (11) and (12) of the mathematical model, Variable y misses its superscript f.

(5) In line 290, ‘4.3’ should be ‘4.2’, and the second ‘?o’ should be ‘ro’.

(6) In Figure 10(a), ‘8-->7’ should be ‘7-->8’ .

(7) In line 442, ‘E2_A, E2_A, and E3_A’ should be ‘E1_A, E2_A, and E3_A’.

(8) In line 465, ‘Section 3.1’ should be ‘Section 5.1’.

Author Response

Dear reviewer,

Thank you very much for your precious reviewing comments. We revised our manuscript carefully based on all of your reviewing comments. Attached is the point-by-point response to the comments. Authors’ responses are in red color font following the comments. Modifications in the manuscript are highlighted in blue color.

Yours sincerely,
Authors

Reviewer 2 Report

Thank you for the opportunity to review this article. This paper studies a vehicle-drone arc routing problem (VD-ARP) consisting of one vehicle and multiple drones.

After reading the paper carefully, my recommendation is a SERIOUS major revision before acceptance for publication.

  1. The abstract should be rewritten seriously by including the major findings and the novelty of the results.
  2. You have not critically reviewed VERY RELEVANT research works to identify a GAP that motivates new research on the exact same topic.
  3. The current form of the Introduction section is short. For example, the authors have not discussed sufficient literature in the introduction. The authors need to highlight the literature gap, major contributions, and motivation. Also describe your major contribution clearly in the introduction.
  4. The presentation and organization of the work MUST be improved. The study requires considerable attention concerning writing style, grammar, and punctuation.
  5. The English writing and linguistic quality of the paper should be improved with the help of an English native speaker.
  6. Contribution of the paper: Make more clear the contribution of the paper by focusing on the improvement of existing methods.
  7. The conclusion writing should be improved carefully. Summaries the advantages and limitations of the proposed method in practical applications. There are many limitations to this study that are not included in the last section.
  8. Generalization of the results: Discuss on the generalization of the results of the study. Is it possible the extension of research findings and conclusions from the study conducted?

Author Response

(The authors gave the same response as above.)

Reviewer 3 Report

The Abstract of the manuscript-at-hand needs to be delineated in a categorical manner, i.e., it should first introduce the domain succinctly, highlight problems  of the domain, delineate the rationale underlying the problem opted for this particular manuscript, and should then summarize the proposed solution.

Line 81 documents the word 'rarely'. What does the authors imply by the same - has the solution been previously proposed in the Literature or has never been proposed? If proposed, what were the issues in those heuristics that have been addressed in this manuscript?

A Table summarizing the salient features and pros and cons of the Literature presented in Section II (Related Work) is indispensable.

The Assumptions presented in Section 3, i.e., Problem Description, are slightly ambiguous, i.e., what's the rationale for assuming that the 'power consumption rate during the drone flight is constant - Line 169' as the same is not realistic by any means. Also, why 'the time of launching/recovering a drone is negligible (Line 172)'?. Putting so many assumptions diminishes the significance of the approach at times. 

Moreover, what would transpire if the Speed of the Vehicles depicted in Table 2 (Experimental Parameter Design) is increased significantly, i.e., vehicles in the context of the Smart Cities would not traverse at mere 30 km/h?

Author Response

(The authors gave the same response as above.)

Round 2

Reviewer 1 Report

In the revised manuscript, the author carefully and accurately responded to my previous comments. Now, I have no other comments. I suggest that the editor can accept this paper.

Reviewer 2 Report

This paper can be accept in this current form.

Reviewer 3 Report

Thank you for addressing the suggested Recommendations.